# Multiple Bone Destruction Secondary to *Mycobacterium kansasii* Pulmonary Disease: A Case Report

**DOI:** 10.3390/diagnostics13111970

**Published:** 2023-06-05

**Authors:** Lu Dai, Yanyan Wu, Xi Zhou, Sen Liu, Junping Fan, Huaiya Xie, Luo Wang, Xinlun Tian, Wenbing Xu

**Affiliations:** 1Department of Respiratory and Critical Care Medicine, Peking Union Medical College Hospital, Chinese Academy of Medical Sciences & Peking Union Medical College, No. 1 Shuaifuyuan Wangfujing, Dongcheng District, Beijing 100730, China; 2Department of Internal Medicine, Peking Union Medical College Hospital, Chinese Academy of Medical Sciences & Peking Union Medical College, No. 1 Shuaifuyuan Wangfujing, Dongcheng District, Beijing 100730, China; 3Department of Orthopaedic Surgery, Peking Union Medical College Hospital, Chinese Academy of Medical Sciences & Peking Union Medical College, Beijing 100730, China

**Keywords:** *Mycobacterium kansasii*, bone destruction, immunocompetent

## Abstract

*Mycobacterium kansasii* infections predominantly manifest in immunocompromised people and are primarily responsible for lung disease and systemic disseminated infection. Osteopathy is a rare consequence of *M. kansasii* infection. Here, we present imaging data from a 44-year-old immunocompetent Chinese woman diagnosed with multiple bone destruction, particularly of the spine, secondary to *M. kansasii* pulmonary disease, which is easily misdiagnosed. The patient underwent an emergency operation after experiencing unexpected incomplete paraplegia during hospitalization, indicating an aggravation of bone destruction. Preoperative sputum testing and next-generation sequencing of DNA and RNA of intraoperative samples confirmed the diagnosis of *M. kansasii* infection. Treatment with anti-tuberculosis therapy and the subsequent patient response supported our diagnosis. Given the rarity of osteopathy secondary to *M. kansasii* infection in immunocompetent individuals, our case offers some insight into this diagnosis.

*Mycobacterium kansasii* is one of the most prevalent species worldwide [1,2], typically found in municipal water [3]; it predominantly affects men and the elderly [4] and has risk factors such as structural lung damage, immunosuppression, and association with certain medications [5]. *M. kansasii* is primarily responsible for lung disease and systemic disseminated infection in immunocompromised individuals and is rare in immunocompetent people. Osteopathy is a rare sequela of such *M. kansasii* infections that commonly manifest as osteomyelitis. Only seven such cases involving immunocompetent individuals have been documented in the literature, of which four involving the spine are identical to our case [6,7,8,9,10,11,12]. Here, we describe a rare case of an immunocompetent patient who had a diagnosed systemic *M. kansasii* infection with spinal involvement and show her imaging performance before (Figure 1) and after (Figure 2) admission.

Although the pathogen development resembles that of tuberculous bacteria, the mechanism of non-tuberculous *Mycobacteria* (NTM) remains unknown. Patients without HIV infections may be affected by anomalies in the IFN–interleukin-12 (IL-12) axis and T cell + lymphopenia [13,14]. In this case, the patient had reduced T-cell and elevated IFN–autoantibody counts. Given the onset of her symptoms after age 40 and no previous history of recurrent infection, a genetic cause for her disease is less likely. However, patients with NTM infections may require testing for latent immunodeficiency, particularly if they exhibit systemic dissemination and a confirmed *M. kansasii* infection, as in this case. This patient was previously suspected and misdiagnosed as having lung cancer with bony metastases in the long term because of the resembling imaging performances of the lung and multiple bone lesions and a family history of lymphoma contributing to her father’s death. This misdiagnosis is clinically common if species identification results are not obtained [15,16]. Therefore, clinicians must look for signs of pathogens while diagnosing cancer. Infections by unique pathogens should be considered for people who exhibit symptoms similar to this patient. The cultivation of NTM should be a part of the ongoing research of potential diseases. Moreover, this patient underwent several invasive procedures and biopsies without conclusive proof, which resulted in persistent complaints and unsatisfactory therapeutic outcomes. Disseminated diseases require widespread diagnostic evidence, and thus, clinicians must acquire specimens from all possible lesions in addition to the lungs. Large tissue samples are warranted when repeated puncture cytology specimens fail to provide a definitive diagnosis. In addition to examining the pathology of the lesions, a pathogen examination should be performed to help with the diagnosis.

## Figures and Tables

**Figure 1 diagnostics-13-01970-f001:**
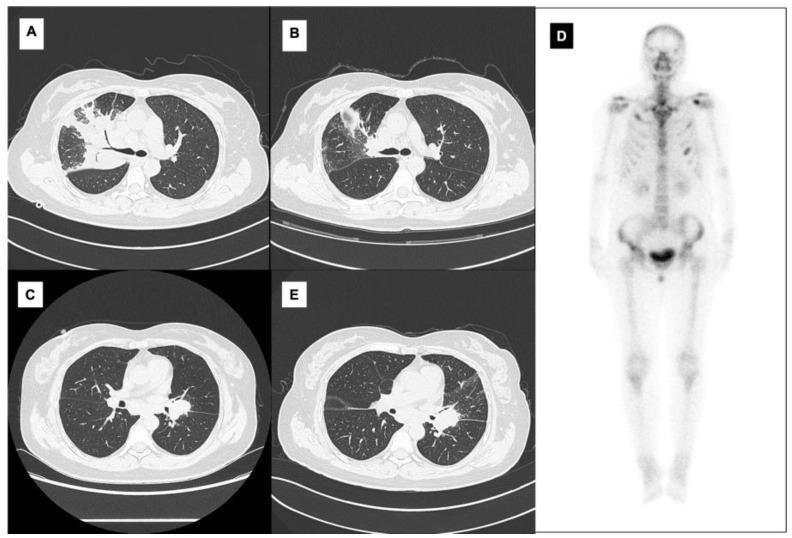
A 44-year-old female presented with a 15-month history of repetitive fever, breathlessness, bone pain, and a 3-month history of hemoptysis. At the beginning of the disease course, chest computed tomography (CT) revealed soft tissue masses with a maximal size of 7.5 cm × 7.2 cm (**A**) in the upper lobe and hilum of the right lung together with local bronchial stenosis and multiple lymphadenectases. Histopathology accessed through transbronchial lung and lymph node biopsies indicated chronic inflammation and no evidence of a tumor. Partially effective results were observed after nearly a month of antibiotic treatment using cefoxitin, Tienam, and Tazocin combined with a 2-day course of methylprednisolone and hydroxychloroquine (**B**). However, the condition was exacerbated following a cold six months before hospitalization, with a new complaint of multiple bone pain. The chest CT reported soft-tissue masses in the hilum of the left lung (3.3 cm × 1.9 cm) (**C**) and bone destruction in a few thoracic and lumbar vertebrae. The bone scan (**D**) and the positron emission tomography (PET)–CT showed multiple lesions of increased radioactivity in the left lung hilar mass, mediastinal lymph nodes, left pleura, the right lower part of the neck, and uterine fundus. Three transbronchial biopsies and an endobronchial ultrasound-guided transbronchial needle aspiration (EBUS–TBNA) revealed chronic inflammation and granulation tissue formation with no evidence of tumor. The cefmetazole–piperacillin sodium combination and sulbactam sodium had a minimal impact on the patient’s response with no significant progress (**E**). She visited another hospital three months before admission with recurrent fever, bone pain, and a new complaint of hemoptysis. Chest CT revealed a progressive left lung hilar tumor, unresponsive to linezolid, Sulperazone, fluconazole, and methylprednisolone.

**Figure 2 diagnostics-13-01970-f002:**
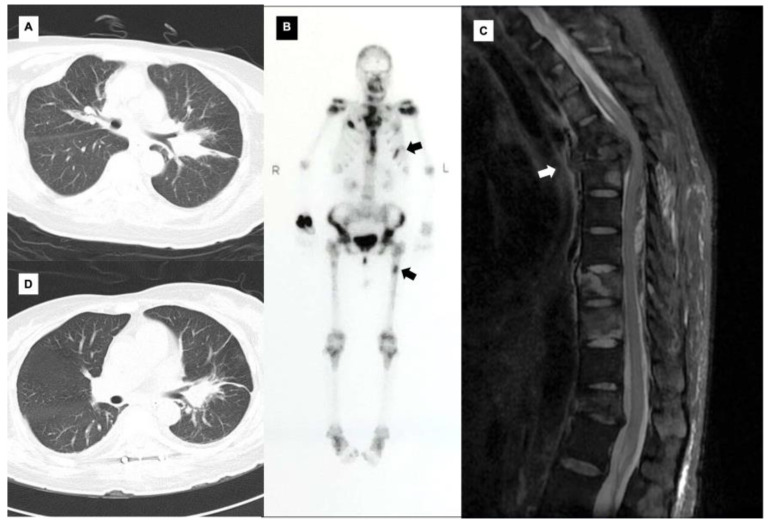
The post-admission chest CT revealed soft tissue in the left lung hilum, multiple lymph nodes, multiple bone destruction, and compression fractures at T5 and T12 (**A**). The bone scan showed various lesions with increased radioactivity in the sternum, numerous vertebrae, and bilaterally in the sternoclavicular joints, multiple ribs, sacroiliac joints, acetabular areas, ischia, shoulder joints, and upper femur. However, the most suspicious lesion of osteomyelitis was found in the left rib and left upper femur (**B**, with black arrows). Laboratory tests showed a high sensitivity C-reactive protein level of 121.38 mg/L (reference range, ≤8.0 mg/L) and a sedimentation rate of 71 mm/h (reference range, 0–20 mm/h). The human immunodeficiency virus (HIV) test was negative. The CD4+ T lymphocyte count was 283 cells/μL (reference range, 561–1137 cells/μL). The anti-interferon-γ (IFN-γ) autoantibodies were concentrated at 1967 (reference range, <20). The nucleic acid amplification test of non-tuberculous *Mycobacteria* (NTM) was positive in sputum, and *Mycobacteria kansasii* was confirmed via polymerase chain reaction (PCR) analysis after two weeks. Until PCR confirmation, the patient presented with uroschesis, hypoesthesia in both lower extremities, and positive reflex of Babinski and Chaddock signs in the right leg. Magnetic resonance imaging of the thoracic vertebrae revealed the centrum and intervertebral disc destruction at the T4 level, with soft tissue invading the spinal canal and compressing the spinal cord (**C**, with a white arrow). After urgent consultation, the patient underwent an emergency posterior thoracic spinal canal decompression at T4–T6; the mass at T5 was partially resected, and internal fixation was performed. Extensive jelly-like purulent necrotic tissue was found in the epidural spinal canal at the T5 level. The pus and vertebral tissue tested positive for acid-fast staining, the cultivation for mycobacterium was positive as well, and *M. kansasii* was further confirmed via next-generation sequencing of DNA and RNA. The patient was thus treated with clarithromycin 0.5 g every 12 h and daily doses of isoniazid 0.3 g, ethambutol 0.75 g, and ofloxacin 0.5 g. Before hospital discharge and throughout the follow-up, the pulmonary and skeletal lesions had improved, and the muscle strength and sensation of the lower extremities were restored (**D**).

## Data Availability

Not applicable.

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
