# Peer review of "Multiple Bone Destruction Secondary to Mycobacterium kansasii Pulmonary Disease: A Case Report"

_diagnostics, 2023, doi:10.3390/diagnostics13111970_

Round 1

Reviewer 1 Report

Given the rarity of osteopathy secondary to M. kansasii infection in immunocompetent individuals, presented case offers some insight into this diagnosis.

Author Response

Thank you for your kind comments. 

Reviewer 2 Report

Despite the topic enclosed in the report perfectly fits with the scope of the journal in the section of Interesting Images, the report should be improved since the presentation is not good. Considering the other articles enclosing the interesting image in Diagnostic, authors should improve the quality of the presentation. All the necessary info cannot be enclosed in the figure captions. Accordingly, the paper should be restructured, enclosing a brief introduction and the caption should be shortened, while the findings with the description of the images should be reported as main text. Furthermore the importance of the findings should be clearly highlighted. After this restyling the paper can be reevaluated.

The english grammar will be evaluated after the revision of the paper

Author Response

Thank you for your comments. Since the article type is "Interesting Images", we formatted our manuscript according to the author's instruction from the Diagnostics editorial office that "no regular manuscript text (introduction/methods/results/discussion) should be included. Instead, images should be accompanied by detailed legends with no restriction in length. Reference citations should appear in the legends.". We consulted the Editor and carefully revised the format in response to your suggestions. We added a brief introduction at the beginning of the manuscript and emphasized the significance of the findings at the end. The descriptions of the images, laboratory tests, and disease processes remained in the captions. We appreciate your understanding and patience.

Reviewer 3 Report

Manuscript presents imaging data from a 44-year-old immunocompetent Chinese woman diagnosed with multiple bone destruction, particularly of the spine, secondary to M. kansasii pulmonary disease, which is easily misdiagnosed. Data presented clearly show pulmonary disease and bone destruction, however ther is no probe for M. kansassii in intraoperative samples, except for comment of the next-generation sequencing of DNA and RNA. As the author stated: "The cultivation of NTM should be a part of the ongoing research of potential diseases" "Disseminated diseases require widespread diagnostic evidence, and thus clinicians must acquire specimens from all possible lesions in addition to the lungs", but this is not the case in the present study and it should.

English is reasonably written and only needs minor style corrections

Author Response

Thank you for this helpful suggestion. We performed acid-fast staining and mycobacterium cultivation tests on intraoperative samples after the operation. At the same time, we requested that the laboratory complete the next-generation sequencing (NGS) of the intraoperative samples' DNA and RNA. The confirmed result of M. kansasii was obtained the next day via NGS, while the positive result of mycobacterial cultivation was obtained 17 days (423 hours) later. We started treatment by combining the results of the PCR analysis for sputum that reconfirmed the diagnosis of M. kansasii (2 weeks after we sent the sputum sample) that came before the results of the intraoperative mycobacterium cultivation. Because of the time required and the known outcome, we stopped making further probes for M. kansasii in intraoperative samples in our laboratory. Despite this, we continued to investigate whether she had latent immunodeficiency due to the disseminated disease caused by M. kansasii. The HIV test was negative, the CD4+ T lymphocyte count was decreased (283 cells/μL, reference range, 561-1137 cells/μL), and anti-interferon- (IFN-) autoantibodies were concentrated in 1967 (reference range, <20), which was elevated but did not meet the standard to confirm immunodeficiency. We did not consider her disease to have a genetic cause because her symptoms began after the age of 40 and she had no prior history of recurrent infection. The results of mycobacterium cultivation in intraoperative samples have been updated in the manuscript.

Round 2

Reviewer 2 Report

The current version of the paper has been improved and now it is suitable for publication in Diagnostics as Interesting Images type of paper. Please address the points below:

-rows 45 and 102 M. kansasi should be in italic

-define NTM at row 96

English language is good.

Author Response

Thank you for pointing out the problems in the manuscript. The manuscript has been revised as follows:

- "M. kansasii" was italicized in rows 45 and 105.

-NTM was defined in the legend (row 83) as well as the main text (row 99).

Reviewer 3 Report

The authors have addressed correctly all my requests.

Minor editing of English language required

Author Response

Thank you for your kind comments.